

# Albumin promotes proliferation of G1 arrested serum starved hepatocellular carcinoma cells

Badr Ibrahim[1], Jan Stange[1], Adrian Dominik[1], Martin Sauer[2], Sandra Doss[1] and Martin Eggert[1]

[1] Division of Nephrology/ Department of Internal Medicine, University Hospital Rostock, Rostock, Mecklenburg Verpommern, Germany
[2] Department of Anaesthesiology and Intensive Care Medicine, University Hospital Rostock, Rostock, Mecklenburg Verpommern, Germany

## ABSTRACT

Albumin is the most abundant plasma protein and functions as a transport molecule that continuously interacts with various cell types. Because of these properties, albumin has been exploited by the pharmaceutical industry to improve drug delivery into target cells. The immediate effects of albumin on cells, however, require further understanding. The cell interacting properties and pharmaceutical applications of albumin incentivises continual research into the immediate effects of albumin on cells. The HepG2/C3A hepatocellular carcinoma cell line is used as a model for studying cancer pathology as well as liver biosynthesis and cellular responses to drugs. Here we investigated the direct effect of purified albumin on HepG2/C3A cell proliferation in the absence of serum, growth factors and other serum originating albumin bound molecules. We observed that the reduced cell counts in serum starved HepG2/C3A cultures were increased by the inclusion of albumin. Cell cycle analysis demonstrated that the percentage of cells in G1 phase during serum starvation was reduced from $86.4 \pm 2.3\%$ to $78.3 \pm 3.2\%$ by the inclusion of albumin whereas the percentage of cells in S phase was increased from $6.5 \pm 1.5\%$ to $14.3 \pm 3.6\%$. A significant reduction in the cell cycle inhibitor protein, P21, accompanied the changes in the proportions of cell cycle phases upon treatment with albumin. We have also observed that the levels of dead cells determined by DNA fragmentation and membrane permeabilization caused by serum starvation (TUNEL: $16.6 \pm 7.2\%$, ethidium bromide: $13.8 \pm 4.8\%$) were not significantly altered by the inclusion of albumin ($11.6 \pm 10.2\%$, ethidium bromide: $16.9 \pm 8.9\%$). Therefore, the increase in cell number was mainly caused by albumin promoting proliferation rather than protection against cell death. These primary findings demonstrate that albumin has immediate effects on HepG2/C3A hepatocellular carcinoma cells. These effects should be taken into consideration when studying the effects of albumin bound drugs or pathological ligands bound to albumin on HepG2/C3A cells.

Corresponding author
Badr Ibrahim,
badr_ibrahim@ymail.com

## INTRODUCTION

Albumin comprises half of the plasma proteins in healthy individuals at concentrations of circa 40 g/L (0.6 mmol/L) and is produced by hepatocytes and exported through the blood to the rest of the cells in the body (*Margarson & Soni, 1998*; *Quinlan, Martin & Evans, 2005*). Albumin is capable of traversing intracellularly between different organs due to its interactions with several cellular receptors. Hence, many pharmaceutical manufacturers bind drugs to albumin in order to improve their circulatory half-life (*Dennis et al., 2002*; *Sleep, Cameron & Evans, 2013*) and delivery into target cells (*Frei, 2011*; *Larsen et al., 2016*)

Albumin's ability to transport between different cell occurs through endocytosis and transcytosis and is controlled by several cellular receptors. These interactions dictate whether albumin should be internalised or cross the vascular endothelial barrier to extravascular compartments (*Schnitzer et al., 1992*; *Vogel et al., 2001*). These cellular receptors are selective when binding albumin based on its ligand profile. This can be exemplified in glycoprotein receptors gp60, gp18 and gp30. Native albumin binds to gp60 and is transported by transcytosis through the endothelial cells whereas modified albumin (by surface adsorption to colloidal Au or maleic anhydride treatment) binds gp18 and gp30 and is internalised by endocytosis to be delivered to lysosomes for degradation (*Schnitzer et al., 1992*; *Schnitzer & Bravo, 1993*). Another study demonstrates neonatal Fc receptor (FcRn) in transporting albumin across endothelial cells. Recombinant albumin was engineered to have high or low affinity to bind FcRn. Albumin with high affinity was recycled whereas low affinity albumin underwent lysosomal degradation (*Schmidt et al., 2017*). These cellular interactions with albumin demonstrate that cells selectively dictate the fate of albumin based on the albumin's ligand conformation. This raises the question of whether the cells in turn respond physiologically to albumin.

Several studies have indeed demonstrated that different cell types not only respond to albumin, but they respond differently. One of the prominent effects of albumin on cells can be demonstrated by albumins role in apoptosis. Albumin has shown to protect against apoptosis during serum starvation in several cell types including endothelial cells (*Zoellner et al., 1996*), Pheochromocytoma cells (*Zhang et al., 2012*) and Neuroblastoma cells (*Gallego-Sandín et al., 2005*). Albumin also protects against ROS activated apoptosis in chronic lymphocytic leukaemia cells (*Moran et al., 2002*) and hybridoma T cells (*Liu et al., 2012*). Albumin can also have a detrimental role, such as that in proximal tubular cells, by causing endoplasmic reticulum stress that consequently leads to apoptosis (*Ohse et al., 2006*). For its beneficiary role, albumin is used as a supplement in serum free media (*Jäger, Lehmann & Friedl, 1988*; *Francis, 2010*) implying its importance in cell maintenance. These effects of albumin however need further interpretation.

It has become evident that the diversity of cellular responses to albumin is not only dependent on the cell type but also on the properties of the interacting albumin. This diversity was demonstrated in rat kidney, human squamous carcinoma and various human neuronal cells having responses to fraction V albumin (HPLC fraction of albumin that contains impurities, mainly fatty acids) that are different from their responses to fatty acid free albumin (*Keenan et al., 1997*; *Hooper, Taylor & Pocock, 2005*). Effects of

albumin on hepatocellular carcinoma cell line have been previously studied demonstrating conflicting outcomes. While one study suggested that albumin stimulates proliferation of hepatocellular carcinoma cells that were inhibited by fatty acids (*Lystad et al., 1994*), other studies suggested hepatocellular carcinoma cells cease proliferation in response to albumin (*Nojiri & Joh, 2014*; *Bağırsakçı et al., 2017*). From the collectively reviewed experiments, it can be extrapolated that the diverse cellular responses to albumin are attributed to the varying albumin receptors in the cell as well as the varying ligand profiles of albumin.

Although, studies demonstrated diverse responses of cellular interactions with albumin, more research is still needed to understand the specifics of these cellular responses. Here we study the effect of albumin on HepG2/C3A hepatocellular carcinoma cell line, a contact inhibited subclone of the HepG2 cell line (*Aden et al., 1979*) that retains some physiological functions of normal hepatocytes (*Kelly et al., 1992*; *Nibourg et al., 2012*). The HepG2/C3A cells are used as a model cell line for studying parenchymal biosynthesis (*Knowles, Howe & Aden, 1980*; *Zannis et al., 1981*; *Nibourg et al., 2012*) and screening for cytotoxicity of drug compounds for side effects involving liver injury (*Gaskell et al., 2016*; *Doß et al., 2017*). The cell line has also been considered for testing clinical samples in screening for diseases (*Sauer et al., 2012*; *Sauer et al., 2018*). Furthermore, the HepG2/C3A cell line has been used in clinical trials as a therapeutic in an extracorporeal bioartificial liver device (*Nibourg et al., 2012*).

Since hepatocytes play an important role in albumin metabolism and albumin bound drug clearance (*Meijer & Van der Sluijs, 1989*), we decided to evaluate the HepG2/C3A cell line's immediate responses to albumin. This simple approach is required to understand basic responses of HepG2/C3A cells to albumin prior to engaging in complex experiments in order to exploit these features for testing albumin bound drugs or effects of different bound albumin profiles on cells. Therefore, we applied charcoal defatted human serum albumin to cultured HepG2/C3A cells in the absence of serum. The immediate responses of the cells, particularly proliferation and cell death, were examined. Although similar studies have been previously performed on the HepG2 hepatocellular carcinoma parent clone demonstrating that albumin prevents proliferation (*Nojiri & Joh, 2014*; *Bağırsakçı et al., 2017*), experiments in this study were carried out on the HepG2/C3A subclone and were contradictory to these earlier findings. Since HepG2/C3A cell line is commonly used as a model for drug testing (*Gaskell et al., 2016*; *Doß et al., 2017*), it is justifiable to have a basic understanding of the effects of albumin on HepG2/C3A cells when testing albumin bound drugs. This primary approach offers a platform and a control method when studying the effects of drug bound albumin and pathologically modified albumin on cell proliferation as well as cytotoxicity.

## MATERIALS & METHODS

### Cell culture

The cell line used in this study is C3A [HepG2/C3A, derivative of Hep G2 (ATCC HB-8065)] (ATCC® CRL-10741™). Cells were routinely cultured in DMEM (Gibco) supplemented with 10% foetal bovine serum (FBS, Sigma) and 2 mM L-glutamine (Biochrom) at a

seeding density of $8 \times 10^4$ cells/cm$^2$ in 25 cm$^2$ flasks (Greiner Bio-one) (Fig. S1). Cells were incubated at 37 °C, 98% humidity and 5% CO$_2$ (BBD 6220 CO$_2$ incubator, Thermo Scientific). Cells were harvested by detachment using trypsin/EDTA, then neutralised with media containing 10% FBS. They were then pelleted by centrifugation at $300 \times$ g, resuspended in media composed of the desired treatments at a total of $2 \times 10^6$ cells $(8 \times 10^4$ cells/cm$^2$). Cell counts were carried out under a light microscope using a Neubauer haemocytometer.

The treatments in this study include serum free media (DMEM with 2 mM L-glutamine, completely without FBS) as the serum starved control, serum free media containing 5 mg/ml albumin (human serum albumin, Octapharma) or serum free media containing 5 mg/ml dextran 70 (Carl Roth). The albumin used in this study is in physiological solution and charcoal treated (Hepalbin, Albutec) to reduce albumin bound stabilisers and other fatty acids prior to applying it to the cells (Chen, 1967). A concentration of 5 mg/ml albumin was applied in our experiments because it is comparable to total protein concentrations in 10% FBS used in routine cell culture.

HepG2 cells typically respond slowly to serum starvation and evidence of cell cycle arrest and apoptosis are usually delayed (Figs. S1 and S2) (Zhuge & Cederbaum, 2006), therefore tests were conducted after 72 h in culture. Cells were routinely viewed using inverted phase contrast microscope (DM IL LED, Leica) and images were captured using a mounted camera (MC120 HD, Leica).

HepG2/C3A cells were authenticated using short tandem repeat analysis and hepatocyte functionality was confirmed by measuring micro albumin synthesis and cytochrome P450 monooxygenase 1A activity. Cells were periodically screened for mycoplasma by extranuclear 4′,6-Diamidine-2′-phenylindole dihydrochloride (DAPI, Carl Roth) staining viewed under ECLIPSE Ti inverted fluorescence microscope (Nikon). Confirmatory testing of mycoplasma enzymatic activity (Mycoalert, Lonza) was carried out according to kit instructions and measured using CLARIOstar plate reader (BMG LABTECH). Both tests have been negative confirming absence of mycoplasma contamination.

## Cell cycle analysis

Harvested cells were fixed in 66% Ethanol at a concentration of $10^6$ cells/ml for a minimum of 2 h at 4 °C. Cells were washed with phosphate buffer saline and incubated with 200 μl staining solution consisting of 50 μg/ml propidium iodide and 550 U/ml RNase A (Abcam) in the dark at 37 °C for 20 to 30 min. Measurements were taken at excitation wavelength of 488 nm using the FACSVerse flow cytometer (BD Biosciences). Gating was carried out using FlowJo 10.5.3 software (FlowJo) and calculated using the software's Watson Pragmatic algorithm to correct for overlaps between the peaks.

## TUNEL assay

Suspended cells were fixed with 1% paraformaldehyde (pH 7.4) for 60 min on ice. The cells were washed with PBS followed by 70% ethanol fixation at −20 °C overnight. Cells were treated with terminal deoxynucleotidyl transferase enzyme (TdT) and fluorescein isothiocyanate (FITC) deoxy uridine triphosphate (FITC-dUTP) provided in the kit

(Phoenix flow systems). Measurements were taken at excitation wavelength of 488nm using the FACSVerse flow cytometer (BD Biosciences).

## Immunoblotting

Cells collected after 72 h treatments were lysed in RIPA buffer (Millipore) containing protease inhibitor cocktail (cOmplete, Roche). Protein concentrations were measured colorimetrically using Pierce BCA protein assay kit (Thermo Scientific). Total protein concentrations of 15 µg for cyclin D1 and 40 µg for p21 in loading buffer (Roti-Load 1, Carl Roth) were separated by SDS-PAGE in a 12% separating gel and 5% stacking gel (Rotiphorese NF-Acrylamide/Bis-solution 30% (29:1), Carl Roth). Proteins were transferred to PVDF membrane (Immobilon P, Millipore) followed by blocking (Roti-Block, Carl Roth), incubation in primary antibody and then secondary antibody. Rabbit monoclonal to p21 [EPR3993] (1:1000, ab109199, Abcam), Rabbit monoclonal to cyclin D1 [EPR2241] (1:3000, ab134175, Abcam) and Rabbit polyclonal to GAPDH (1:1000, ab9485, Abcam) were used as primary antibodies. The secondary antibody used was HRP conjugated Goat F(ab')2 Anti-Rabbit IgG (1:7500, ab6013, Abcam). The membrane was washed between incubations with TBST (20 mM Tris–HCl (pH 7.6), 0.137 M NaCl, 0.05% Tween 20). The blots were then developed using Pierce ECL plus (Thermo Scientific) and imaged using Fusion FX (Vilber). The blots were analyzed densitometrically using ImageJ (https://imagej.nih.gov).

## Fluorescence imaging

Cells were cultured on a 24 well plate (Greiner) at a seeding density of $8 \times 10^4$ cells/cm$^2$ for 72 h with the different treatments. Unfixed adherent cells were directly stained with a mixture of 0.6 µM calcein AM (Invitrogen), 2 µM ethidium bromide (Invitrogen) and 4 µM 4′, 6-Diamidine-2′-phenylindole dihydrochloride (DAPI, Carl Roth) and incubated at 37 °C for 20 min. The cells were viewed and photographed under ECLIPSE Ti inverted fluorescence microscope (Nikon). Cells positive for membrane permeabilization were measured as the proportion of cells stained with ethidium bromide from total cell numbers stained with DAPI. A minimum of 1,000 cells and three fields of view were counted per well.

## Statistics

Four biological replicates were used in each treatment unless stated otherwise. $F$ test was used to determine equal variances. Two tailed $T$-test assuming equal or unequal (Welch test) variances was carried out accordingly to determine statistical significance when comparing two groups. One way-ANOVA followed by Bonferroni's post hoc analysis was carried out when comparing multiple groups against the control. $P$ value of less than 0.05 is considered significant ($\alpha = 0.05$).

## RESULTS

### Albumin alters cell morphology and results in increased cell counts of serum starved HEPG2/C3A cells

To test the immediate effects of albumin, cells were cultured in serum free media containing 5mg/ml albumin and compared with the cells cultured in serum free media without albumin (serum starved control). Cells cultured in serum free media containing 5 mg/ml dextran, with a comparable molecular weight to albumin (70 kDa), was carried out as a control for effects that might relate to oncotic pressure.

Serum starved controls displayed a slight irregular morphology but retained the epithelial polygonal shape and remained as monolayers in colonies. Noticeable morphological differences were observed in albumin treated cells when compared to the serum starved control. Cells remained in colonies but displayed a rounded morphology, formed clusters and grew in more than one layer. Dextran treated cells demonstrated a morphological effect more in resemblance to serum starved cells and did not exhibit the rounded morphology, clusters and layering seen in albumin treated cells (Fig. 1).

The effects of albumin on total cell counts of serum starved cells was evaluated in comparison to Dextran treated cells and serum starved control. Data passed equal variance test and were analysed by one-way ANOVA followed by Bonferroni's multiple comparison versus control (serum starved cells). Serum starvation for 72 h yielded an average total cell count of $2.8 \times 10^6$ cells. Inclusion of albumin in serum starved cells cultured for 72 h significantly increased the average total cell count to $4 \times 10^6$ cells ($p = 0.019$, $n = 4$, $\alpha = 0.05$). Dextran treatment of serum starved cells had fewer cells ($2 \times 10^6$ cells) than the serum starved control but the difference was not significant ($p = 0.14$, $n = 4$, $\alpha = 0.05$) (Fig. 1). These findings suggest that albumin alters cell morphology and results in increased total cell counts in serum starved cells. These effects are not simply due to osmotic pressure and are specific for albumin as these results could not be reproduced in cultures treated with an equal molecular weight of Dextran. Serum starved HepG2/C3a cells demonstrated a similar response of increased total cell counts when higher doses of albumin (25 mg/ml and 50 mg/ml) were included in the cultures (Fig. S3).

### Albumin promotes proliferation but does not prevent cell death in serum starved HEPG2/C3A cells

Albumin triggered increased cell counts in serum starved HEPG2/C3A cells requires further explanation into whether these effects are a result of promoting proliferation or preventing cell loss. HEPG2/C3A cells cultured in media containing 10% FBS were maintained for 120 h until confluent (Fig. S4) and possibly contact inhibited (*Kelly et al., 1992*; *Davis, Ho & Dowdy, 2001*; *Cho et al., 2005*). Cell cycle analysis demonstrated that the cells cultured in 10% FBS for 120 h were 86.6% in G1 phase, 6.2% in S phase and 7.2% in G2/M phase (Fig. S4). The confluent cells were harvested then seeded at the regular seeding density ($2 \times 10^6$ cells) in serum starved media and serum starved media containing 5mg/ml albumin. Cell cycle analysis after 72 h in culture demonstrated a significant reduction in G1 phase in albumin treated cells ($78.3 \pm 3.2\%$) compared to serum starved cells ($86.4 \pm 2.3\%$) (equal variance $t$.test, $p = 0.006$, $n = 4$, $\alpha = 0.05$). Whereas S phase was significantly higher

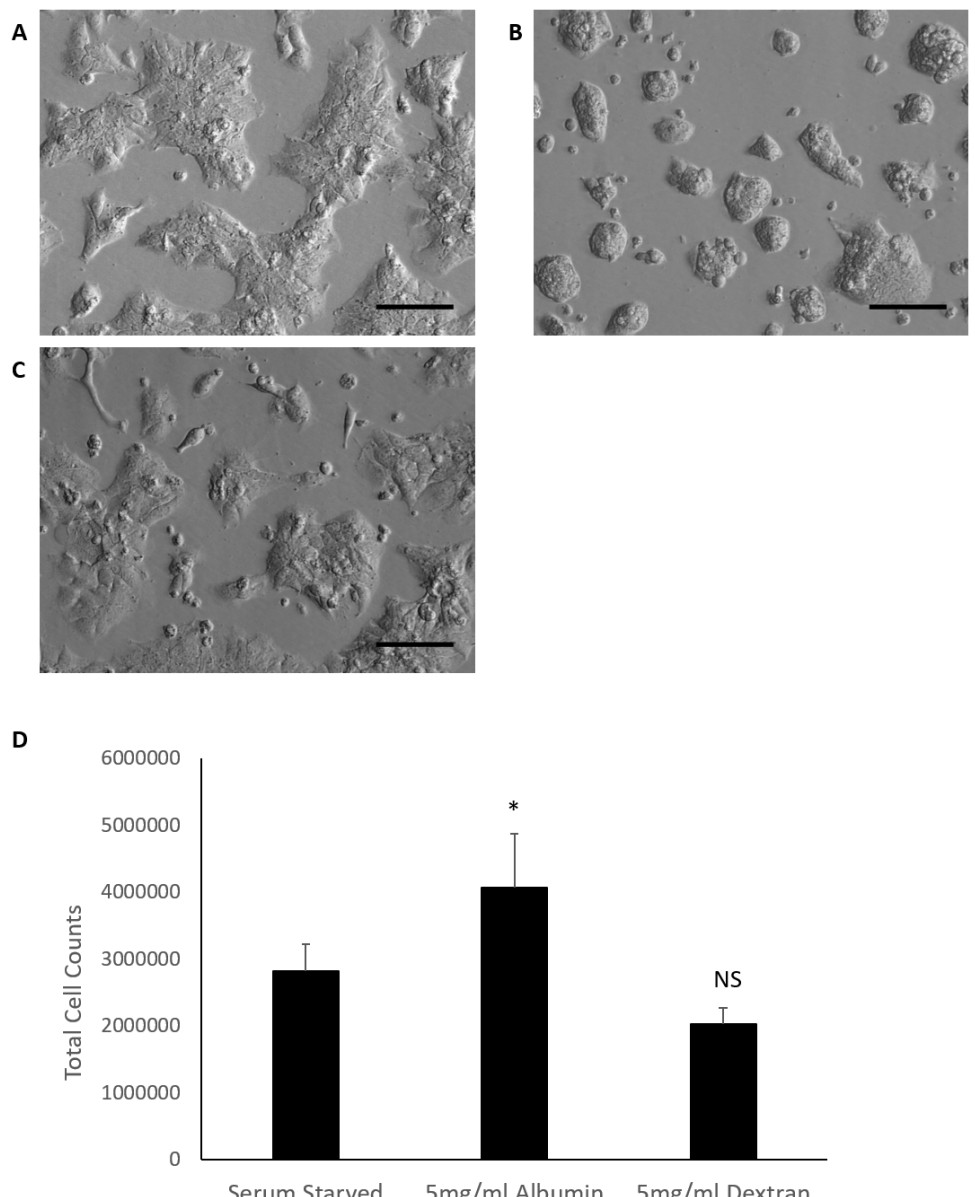

**Figure 1** **Comparison of HepG2/C3A cell morphology and total cell counts after 72 h treatments.** Albumin alters morphology of serum starved HepG2/C3A cells demonstrated in micrographs (20× objective lens) of cells cultured for 72 h in (A) serum free media (serum starved control), (B) serum free media containing 5 mg/ml albumin and (C) serum free media containing 5 mg/ml dextran. Scale bar = 100 μm. (D) Albumin results in increased total cell counts after 72 h in culture. Values are mean ± SD ($n = 4$). *$p < 0.05$, NS= not significant.

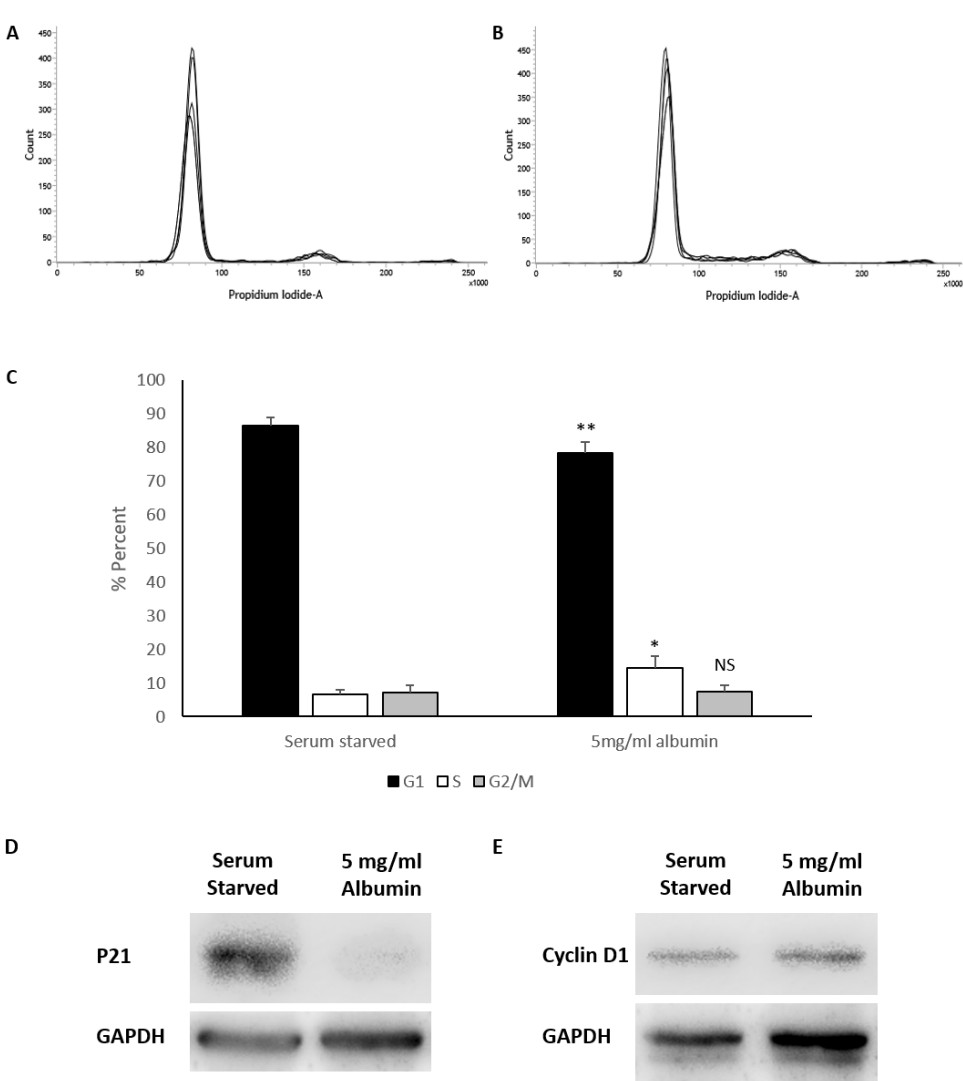

**Figure 2  Cell cycle analysis of serum starved cells with or without albumin.** Flow cytometry histograms of propidium iodide stained cells treated for 72 h with (A) serum starved media or (B) serum starved media containing 5 mg/ml albumin. (C) Percentage of cell cycle stages calculated from the histogram using the Watson pragmatic algorithm shows a decrease in G1 phase and an increase in S phase in albumin containing media. Values are mean ± SD ($n = 4$). *$p < 0.05$, **$p < 0.01$, NS= not significant. Representative images of western blots demonstrating significantly reduced (D) P21 levels but no significant change in (E) cyclin D1 levels in albumin containing cultures compared to serum free controls.

in albumin treated cells (14.3 ± 3.6%) compared to serum starved cells (6.5 ± 1.5%) (unequal variance (Welch) $t$.test, $p = 0.016$, $n = 4$, $\alpha = 0.05$). There was however no significant difference in G2/M phase between albumin treated cells (7.4 ± 1.9%) and serum starved cells (7 ± 2.2%) (equal variance $t$.test, $p = 0.8$, $n = 4$, $\alpha = 0.05$) (Fig. 2 and Fig. S1). These results suggest that serum starvation resulted in a cell cycle arrest in G1 phase, while inclusion of albumin promoted cell cycle transition into S phase.

Western blot analysis demonstrates that levels of p21 (average relative density) expressed in serum starved cells ($1 \pm 0.64$) is significantly reduced (equal variance $t$.test, $p = 0.034$, $n = 4$, $\alpha = 0.05$) by the inclusion of 5 mg/ml albumin ($0.12 \pm 0.08$). Levels of cyclin D1 between serum starved cells ($1 \pm 0.72$) and albumin treated cells ($1.07 \pm 0.87$) were not significantly different (equal variance $t$.test, $p = 0.0$, $n = 3$, $\alpha = 0.05$) (Fig. 2). This suggests that albumin promotes cell cycle progression in serum starved cells by suppressing p21.

Serum starvation causes cell apoptosis and necrosis in cultured cells. The effects of serum starvation are however slow in HepG2 cells (*Zhuge & Cederbaum, 2006*; *Liang et al., 2013*). This slow response is recapitulated in HEPG2/C3A cells (Figs. S2 and S5); hence, the 72-hour time point was selected for this study. Apoptosis and necrosis were measured in HEPG2/C3A cells by TUNEL assay and ethidium bromide staining respectively. Serum starved cells were $16.6 \pm 7.2\%$ apoptotic and albumin treated cells were $11.6 \pm 10.2\%$ apoptotic (Fig. 3). Although TUNEL positive cells were perceptibly lower in some of the samples when treated with albumin (Fig. S6), the effect was not statistically significant (unequal variance (Welch) $t$.test, $n = 4$, $p = 0.46$, $\alpha = 0.05$) (Fig. 3).

Cells in culture for 72 h were stained with calcein AM (green) and ethidium bromide (red) to demonstrate presence of live and necrotic cells respectively (*Decherchi, Cochard & Gauthier, 1997*). DAPI (blue) nuclear stain was carried out to quantify total number of cells in the field of view. Live cells were characterised by esterase activity that causes calcein AM to fluoresce in the cytoplasm and is clearly present in all treatments. Necrotic cells were characterised by damaged cell membrane that is permeable to ethidium bromide allowing it to stain the nucleus (Fig. 4). Serum starvation resulted in $13.8 \pm 4.8\%$ necrosis. Inclusion of 5mg/ml albumin resulted in an increase in necrosis to $16.9 \pm 8.9\%$ but is not significantly higher than serum starved cultures (equal variance $t$.test, $n = 4$, $p = 0.5$, $\alpha = 0.05$) (Fig. 4). Cells grown in 10% FBS were used as negative controls for TUNEL assay ($0.8 \pm 0.2\%$ apoptotic) and ethidium bromide staining ($1.7 \pm 0.8\%$ necrotic) (Fig. S7). While inclusion of albumin significantly accounts for an increase in cell counts and proportion of cells in S phase during serum starvation it does not significantly protect against cell death by either apoptosis or necrosis.

## DISCUSSION AND CONCLUSIONS

The frequent interactions of naturally abundant albumin with cells in the body as well as its wide exogenous applications in the biomedical and pharmaceutical industries (*Dennis et al., 2002*; *Sleep, Cameron & Evans, 2013*) prompted us to study the effects of albumin on cells. We have selected HepG2/C3A carcinoma cells because they originate from hepatocytes (*Aden et al., 1979*) and have been used as a model for studying hepatocellular carcinoma (*Chen et al., 2015*; *Ao et al., 2017*) as well as normal hepatocyte functions (*Nibourg et al., 2012*; *Gaskell et al., 2016*). Hepatocytes play an important role in the clearance of albumin bound substances (*Meijer & Van der Sluijs, 1989*), therefore some of these albumin interacting properties can potentially be used in targeting hepatocellular carcinoma or studying the effects of pathologically modified albumin on cellular function. This study aims to determine immediate effects of albumin on HepG2/C3A cell proliferation and

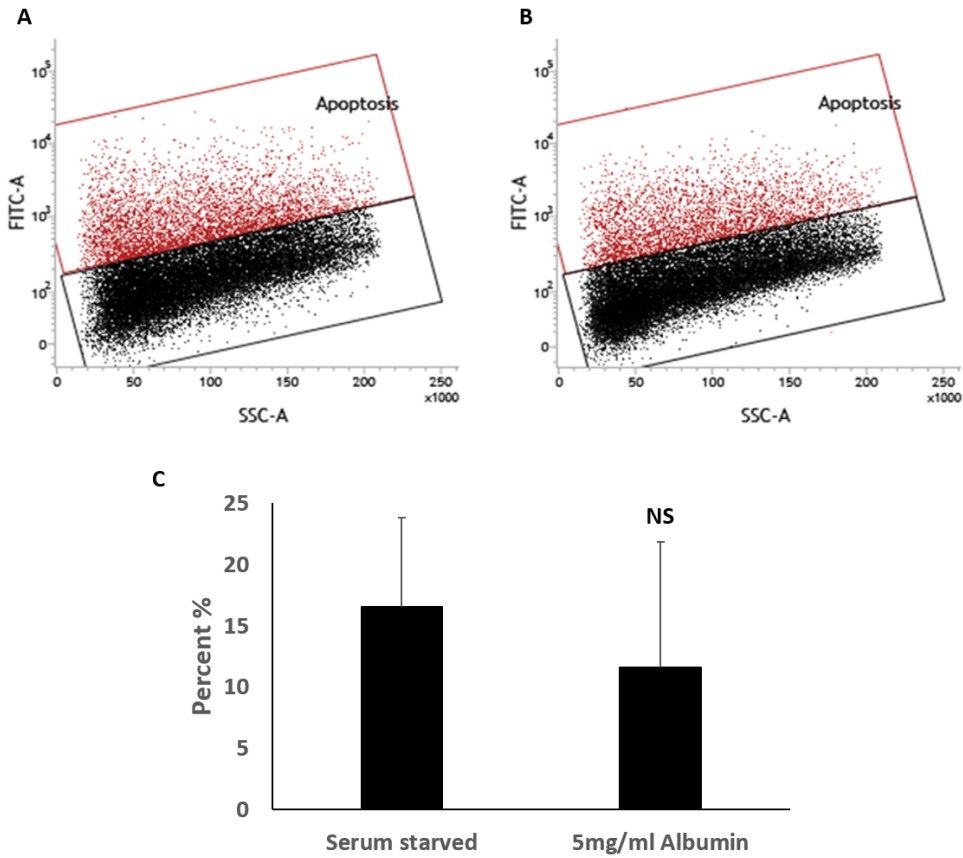

**Figure 3** **TUNEL assay for the measurement of percentage of cells with fragmented DNA.** Percentages of apoptotic cells were not significantly different demonstrated in flowcytometry overlay of dot plots of TUNEL positive cells (red) and negative cells (black) of 72 h cultures of HepG2/C3A in (A) serum starved media or (B) serum starved media containing 5 mg/ml albumin. (C) Bar chart demonstrating percentage of TUNEL positive cells. Values are mean $\pm$ SD ($n = 4$). NS, not significant.

death during serum starvation. Here the effects of albumin free from ligands is carried out to pave the way for understanding how albumin bound ligands may allosterically affect the cell. There have been several studies (see introduction) describing the receptor interactions of a variety of cells with albumin dependent on the albumin's ligand profile and how these cells determine the fate of this albumin molecule dependent on its ligand profile. Most of these studies describe the effects of cells on albumin but studies on the effects of albumin on cells are lacking. Hence, we carried out these experiments.

Although, the effects of albumin on the proliferation of hepatocellular carcinoma cells have been previously studied, the outcomes of those studies are debatable. They argued that albumin inhibits tumour evidenced by its increased levels correlating with cancer remission and addition of albumin to serum starved hepatoma cells caused G1 arrest (*Nojiri & Joh, 2014*; *Bağırsakçı et al., 2017*). It is worth noting that serum is essential for hepatocellular carcinoma growth (*Zhuge & Cederbaum, 2006*; *Liang et al., 2013*) whereas

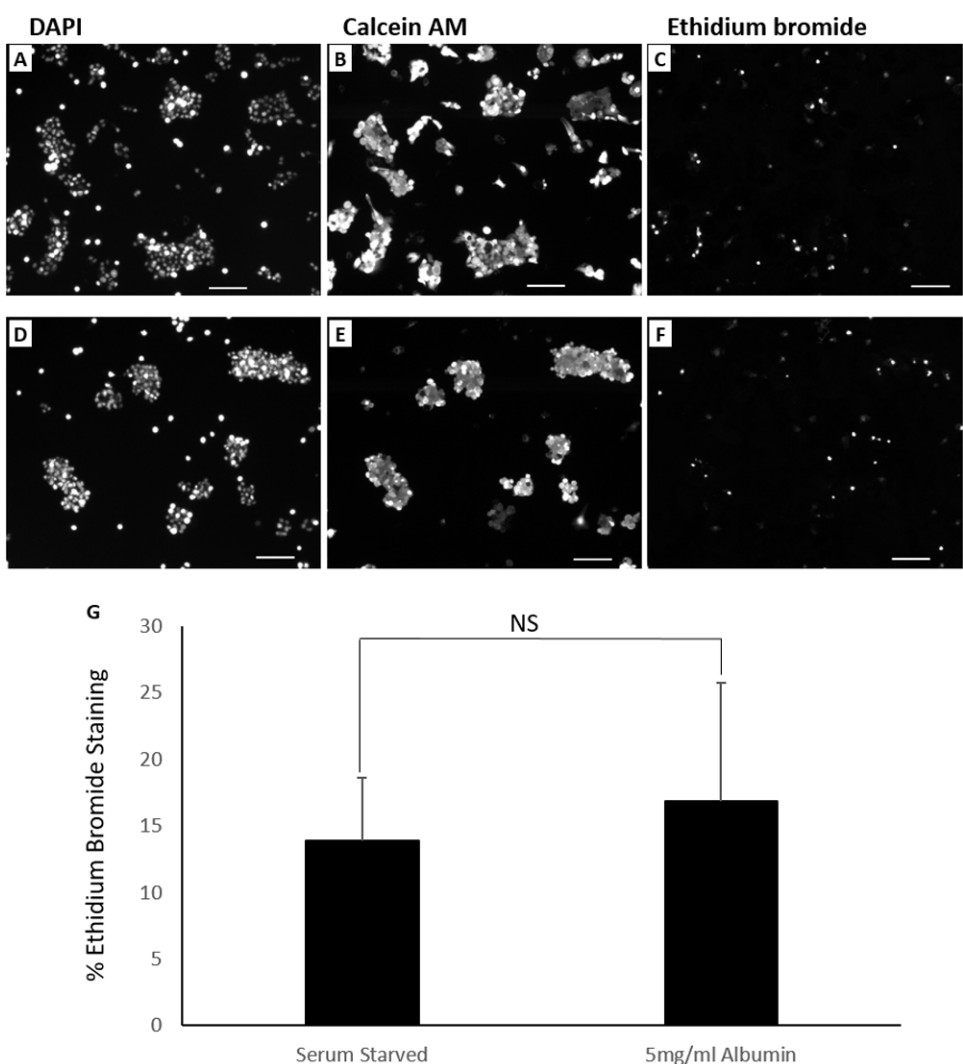

**Figure 4 Live dead fluorescent micrographs of cells.** Percentages of dead cells were not significantly different demonstrated by fluorescence microscopy images of cells (10× objective lens) cultured for 72 h in (A, B, C) serum starved media and (D, E, F) serum starved media containing 5 mg/ml albumin. Fluorochromes used were DAPI (nuclear), calcein AM (cytoplasmic) and ethidium bromide (nuclear). Scale bar = 100 µm. (G) Chart demonstrating differences in the percentage of ethidium bromide stained cells between treatments. Values are mean ± SD ($n = 4$). NS, not significant.

serum starvation is routinely carried out to arrest cells in G1 phase (*Davis, Ho & Dowdy, 2001*; *Langan, Rodgers & Chou, 2017*).

 We carried out our experiments on HEPG2/C3A cells, a selected colony of HepG2 cells, that are responsive to contact inhibition. In our approach, the cells were collected for experimentation after they were grown to confluency to synchronize them by means of contact inhibition (*Kelly et al., 1992*; *Davis, Ho & Dowdy, 2001*; *Cho et al., 2005*). Cells were then grown under serum starved conditions with or without albumin. The initial observed effects of albumin on serum starved HEPG2/C3A cells were morphological. Serum starved

cells retain epithelial morphology and the ability to spread. Albumin caused cells to appear more round, small and clustered (Fig. 1). The cells were then counted and analysed for cell cycle after 72-hour treatments. The reason for selecting this time point is that during serum starvation, earlier time points did not demonstrate sufficient cell cycle arrest whereas at later time points, the cells were mostly dead. At 72 h after serum starvation we observed minimal cell loss (mean number of cells harvested were not lower than cells seeded) and sufficient G1 cell cycle arrest was observed (Fig. S2). With the inclusion of albumin, we observed an increase in cell counts that correlated with a reduction in percentages of cells in G1 phase and an increase in S phase compared to serum starved controls. Furthermore, we investigated levels of cell cycle promoter protein cyclin D1 and cell cycle inhibitor protein p21 (Reed et al., 1994) in serum starved cells and albumin containing cultures. Results demonstrate that inclusion of albumin in serum starved cultures significantly reduces P21 but does not significantly alter levels of Cyclin D1 (Fig. S2). This suggests that albumin permits cell cycle transition from G1 to S phase by downregulating p21 which results in increased cell proliferation.

To further rule out proliferation inhibitory effects of albumin that were suggested in the studies mentioned above (Nojiri & Joh, 2014; Bağırsakçı et al., 2017), we treated HEPG2/C3A cells grown in media containing FBS with 10mg/ml albumin and did not observe a significant change in cell counts (equal variance $t$.test, $n = 3$, $p = 0.36$) (Fig. S8).

Cell death was measured to rule out a protective effect of albumin. TUNEL assay was carried out to determine cells that underwent DNA fragmentation, an indicator of apoptosis (Crowley, Marfell & Waterhouse, 2016). Nuclear staining of cells with ethidium bromide was carried out to demonstrate membrane damage as a result of necrosis or secondary necrosis (Zong & Thompson, 2006). Inclusion of albumin in serum starved cells did not significantly change the proportion of dead cells identified by these methods. This suggest that albumin does not protect against DNA fragmentation and membrane permeabilization that occurs during serum starvation.

We have argued that albumin has an immediate effect on cell proliferation by promoting cell cycle transition from G1 to S phase in the absence of serum and other proteins including growth promoting factors. However, we did not study whether albumin directly interfered with molecular pathways of transcription factors involved in cell cycle transition or indirectly influenced proliferation through interfering with other processes such as cell spreading. A deeper understanding of the mechanism of albumin's effect on cell proliferation requires more detailed studies. This can include identification of receptors and downstream signalling pathways triggered by albumin-receptor interactions.

We investigated modes of cell death individually, but we did not examine any overlaps between DNA fragmentation and membrane permeabilization. Furthermore, we did not examine whether the underlying cause of death in the membrane permeabilized cells was apoptosis or direct necrosis. The tests simply suggest that albumin did not prevent various fates of cell death that occur during serum starvation.

These findings argue that the addition of albumin resulted in increased cell counts as a result of increased proliferation through promotion of G1 to S phase transition and not by prevention of cell death in serum starved HepG2/C3A cells. This study offers

primary results and a platform for further investigations into the molecular interactions of albumin with cells. Additionally, this approach can be used as a control to study different modifications of albumin, ligand profiles and drug bound albumin.

## ACKNOWLEDGEMENTS

We would like to thank Heike Potschka (Fraunhofer Institute for Cell therapy and Immunology, Rostock, Germany) for maintaining a standard cell culture laboratory, without which we would not have been able to proceed with our experiments. We would like to thank Wendy Bergman (Core Facility for Cell Sorting and Cell Analysis, University Medical Centre Rostock, Rostock, Germany) for her support with optimising the flow cytometry experiments.

### Funding

This study was financed by the state of Mecklenburg Vorpommern with grants from the European Regional Development Fund (ERDF). (Grant reference number: TBI-1-110-VBW-038). The funders had no role in study design, data collection and analysis, decision to publish, or preparation of the manuscript.

### Grant Disclosures

The following grant information was disclosed by the authors:
European Regional Development Fund: TBI-1-110-VBW-038.

### Competing Interests

The authors declare there are no competing interests.

### Author Contributions

- Badr Ibrahim conceived and designed the experiments, performed the experiments, analyzed the data, prepared figures and/or tables, authored or reviewed drafts of the paper, and approved the final draft.
- Jan Stange conceived and designed the experiments, authored or reviewed drafts of the paper, and approved the final draft.
- Adrian Dominik performed the experiments, prepared figures and/or tables, and approved the final draft.
- Martin Sauer analyzed the data, authored or reviewed drafts of the paper, and approved the final draft.
- Sandra Doss performed the experiments, prepared figures and/or tables, and approved the final draft.
- Martin Eggert conceived and designed the experiments, analyzed the data, authored or reviewed drafts of the paper, and approved the final draft.

### Data Availability

   The raw measurements are available in the Supplementary Files.

## Supplemental Information

Supplemental information for this article can be found online at http://dx.doi.org/10.7717/peerj.8568#supplemental-information.

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
