# Peer review of "Albumin promotes proliferation of G1 arrested serum starved hepatocellular carcinoma cells"

_PeerJ, doi:10.7717/peerj.8568_

## Round 0.1 · original submission · Major Revisions

Please attend to the reviewers´ comments on a detailed point-by-point letter with your rebuttal. A testable model for the action of albumin on cell growth. One of the reviewers has come to our attention the fact that Nojiri (2014) found opposite results to the findings reported here.

In particular, please comment and suggest reasons for the results reported by Nojiri (2014)

Is albumin working as a carrier of molecules, peptides, and hormones, and therefore its cell growth capacity will depend on the molecules bound?

Nojiri S, Joh T. 2014. Albumin suppresses human hepatocellular carcinoma proliferation and the 434 cell cycle. International journal of molecular sciences 15:5163–74

Reviewer 1 ·

Basic reporting

The majority of the manuscript is well written and clear. Concerns regarding Discussion are made in General Comments.

Experimental design

Appropriate experiments have been undertaken with correct controls.

Validity of the findings

no comment.

Additional comments

In this study the authors have addressed a simple question regarding the impact of albumin on the proliferation of HepG2/C3A cells. The justification being that albumin is utilised as a drug delivery system and HepG2/C3A cells are used as a model for screening drug cytotoxicity and side effects involving liver injury. Fundamentally, the authors conclude from their preliminary findings that albumin alone may promote cell growth in the absence of serum. The majority of the manuscript is well written and clear. Appropriate experiments have been undertaken with correct controls.

There are however some minor issues which can be corrected to improve the manuscript.

(1) Use of the word “stagnate” in abstract is not appropriate and should be defined more scientifically. Do the authors mean that cells cease to proliferate or proliferation is reduced?
(2) Percentages for the increases in cell growth/cell phases/cell death etc should be provided in the abstract.
(3) Authors may want to reorder columns in graph 1 D so serum starved is first and albumin is second to reflect order in main text.
(4) Are cell counts mentioned total cell counts or cells per ml. It is not clear and should be stated in text and figures.
(5) Figure legends could be improved by stating main finding of figure rather than what was done.
(6) It would help if figure 2a and b were labelled with either starved or albumin. The same is true for all figures.
(7) Whilst there appears to be a small but significant increase in growth after 72hrs with albumin treated cells in figure 2, further time points would be more convincing. Could the authors justify why further time-points were not provided and provide rationale in main text?
(8) Figure 3c required axis to be labelled.
(9) There appears to be no reference to figure 4 in main text. It would help if there was labelling of A and B in X axis of figure 4. A more detailed labelling of x Axis of Figure 4 C is required.
(10) Do the authors think that albumin stimulates G1 arrested cells into proliferation or rather allow a small number of cells that have not yet undergone G1 arrest to continue to proliferate? Maybe a point for discussion.
(11) The Discussion appears to be mostly a repeat of results and methods with less emphasis on discussion. The authors should consider improving discussion to reflect more this section concisely. The Authors should highlight the pitfalls of their preliminary data and suggest what needs to be done further to improve.
(12) The Authors should also clearly state why they think their findings are important, providing specific examples. Is the relevance of their findings geared toward drugs targeting cancer that are actively proliferating rather than normal quiescent cells? The physiological relevance if any of their data could be discussed. Maybe the data is only relevant to in vitro models and drug testing, if so, how?
(13) References are quite extensive for such a short report.

Reviewer 2 ·

Basic reporting

a) The English language should be improved.
Eg: The title of the first result, lines 201-202: “Albumin alters cell morphology and results in increased cell counts in sérum starved HEPG2/C3A cells” could be rewritten as “Albumin alters cell morphology and results in increased of counts serum starved HEPG2/C3A cells”.

b) Some texts present in the introduction should be other sections.
Lines (126-128): “Proliferation was evaluated through changes in cell counts and cell cycle analysis whereas cell death was measured by analysing apoptotic DNA fragmentation and membrane permeabilization”. This text should be in the materials and methods section.
LInes (131-134): “Our results contrarily suggest that albumin promotes proliferation of serum starved HEPG2/C3A hepatocellular carcinoma cells by allowing G1 to S phase cell cycle transition. We also show that albumin does not significantly interfere in cell death by apoptosis or necrosis”. This text should be in the discussion section.

Experimental design

Your study did not show the molecular mechanism of the observed effect and presents an opposite result to a previous study (Nojiri, 2014). In the Nojiri study, the expression of proteins involved in proliferation signaling was modified, such as p21. Thus, it is necessary to include the evaluation of cell proliferation signaling pathways to prove your hypotheses.

The albumin concentration applied was 5mg / mL because it is comparable to total protein concentrations in 10% FBS used in cell culture. However, the plasma concentration in healthy humans is approximately 40mg / mL and other studies such as that of Nojiri, 2014 applied a concentration of 50mg / mL. Thus, it would be interesting to experimentally evaluate different concentrations of albumin, especially those higher concentrations to resemble the concentration in human plasma and determine if the observed effect is dose dependent.

Validity of the findings

The results shown need additional experiments to show by which mechanism albumin enhances cell proliferation.

Additional comments

Research question is well defined and relevant, however the results need to be complemented with the evaluation of cell proliferation signaling pathways.

---

## Round 0.2 · accepted · Accept

The manuscript has improved over the review rounds and it is now accepted at PeerJ.